# Online shopping green product quality supervision strategy with consumer feedback and collusion behavior

Hui He, Lilong Zhu*

School of Business, Shandong Normal University, Ji'nan, Shandong, China

* zhulilong2008@126.com

**Data Availability Statement:** All relevant data are within the manuscript.

**Funding:** This work was supported by the Humanities and Social Sciences Foundation of the Ministry of Education in China under grant

## Abstract

With the development of e-commerce, online shopping has become one of the most important consumer channels. However, the lack of government supervision, insufficient review of e-commerce platforms, illegal sales of online sellers and invalid consumer complaints have led to frequent green product quality problems during online shopping. Therefore, this paper considers that the online seller may be driven by interests, colluding with the e-commerce platform and selling low quality green product. At the same time, we introduce consumer feedback, and take the government supervision department, the online seller and the e-commerce platform as actors of the evolutionary game. In this paper, the evolutionary strategy choices of each actor were analyzed, and the influence of different factors on the evolutionary stability results was explored. Research indicates: firstly, consumer complaints play an indirect regulatory role for the online seller; secondly, the enhancement of the loss-sharing relationship between the online seller and the e-commerce platform can promote the legal operation of the two and prevent collusion; thirdly, the impact of consumer complaints on the choice of the e-commerce platform depends on the government supervision department's penalty for the e-commerce platform; finally, the e-commerce platform establishes a reasonable reward system, which can make up for the defects of the online seller using advanced technology to avoid punishment. Our paper uses Matlab 2017 for simulation analysis and provides effective advices on how to urge the government supervision department to effectively supervise, promote the e-commerce platform to enhance review, urge the online seller to legal sale, and encourage consumers to legally defend their rights.

## Introduction

In recent years, the popularity and promotion of the Internet in China has promoted the transformation of traditional shopping to online shopping. The 43rd "Statistical Report on the Development of China's Internet Network" released by China Internet Network Information Center (CNNIC) [1] shows: As of December 2018, the number of Internet users in China reached 829 million, with a penetration rate of 59.6%. The number of online shopping users reached 610 million, an increase of 14.4% year-on-year, accounting for 73.6% of the netizens.

No.17YJA630147, Nature Science Foundation of Shandong Province under grant No. ZR2019MG017 and National Social Science Foundation of China under grant No.13AGL012. The funders had no role in study design, data collection and analysis, decision to publish, or preparation of the manuscript.

**Competing interests:** The authors have declared that no competing interests exist.

Online shopping market is ushered in a more promising future. At the same time, countries around the world are actively encouraging the development of green products. Green products refer to the production process and its own energy-saving, water-saving, low-pollution, low-toxic, renewable, recyclable products, which can directly promote the change of people's consumption concepts and lifestyles, and aim to protect the environment. The online shopping environment provides a broader space for the green product market, enabling consumers to not only enjoy the convenience of online shopping, but also contribute to the protection of the environment.

However, the online shopping method also causes serious information asymmetry between the online seller and the buyer, which leads to frequent problems in the quality of online shopping products. The 2018 China E-Commerce User Experience and Complaint Monitoring Report [2] shows: In 2018, complaints received by the e-commerce consumer dispute mediation platform increased by 38.36% year-on-year, among which retail e-commerce complaints accounted for the highest proportion of all complaints, reaching 62.55%. Among the national e-commerce complaint hot issues, the quality of the product is second only to the issue of refunds. For green products, the quality of products should be measured from two aspects, one is the environmental quality of the products, and the other is the quality of the products themselves, which undoubtedly increases the difficulty of quality supervision of green products, coupled with the shortcomings of online shopping, the quality supervision of online shopping green products is more difficult to solve.

In August 2018, China officially promulgated the "Electronic Commerce Law of the People's Republic of China". The promulgation of this law fully reflects the state's high attention to the field of e-commerce, and also reflects the growing problems in the development of e-commerce, which need to be standardized and resolved. On July 16, 2019, the "Guiding Opinions of the General Office of the State Council on Accelerating the Construction of a Social Credit System to Build a New Credit-Based Supervision Mechanism" issued by the General Office of the State Council pointed out that credit is the key point to achieve full chain supervision. The new regulatory mechanism has far-reaching implications for promoting high quality development, and it also places higher demands on the main players. In response to the issue of green products, on May 5, 2019, the General Administration of Market Supervision formulated the "Administrative Measures on the Use of Green Product Labels". The purpose is to implement the relevant tasks of the "Overall Plan for the Reform of Ecological Civilization System" and the "Opinions of the General Office of the State Council on Establishing a Unified Green Product Standard, Certification, Labeling System" issued by the Central Committee of the Communist Party of China and the State Council and to promote the orderly development of green products.

With the continuous deepening of the supply-side structural reforms, the e-commerce environment is required to be more standardized, and the public's awareness of rights protection and environmental awareness are increasing, and more and more appeal to high quality green products. How to strengthen the quality supervision of online shopping green products, protect the legitimate rights and interests of consumers, caused the government supervision departments, e-commerce platforms, scholars and consumers to think deeply. The quality supervision of online shopping green products involves multiple stakeholders, and the information asymmetry in the e-commerce environment makes it difficult to purify the market environment by relying solely on the supervision of government departments. Therefore, this paper introduces the multi-party entities such as the government supervision department, the e-commerce platform, the online seller and consumer closely related to the quality supervision of online shopping green products into the evolutionary game model. Under the consideration of the collusion behavior between the e-commerce platform and the online seller, the

evolutionary stability strategies of each player are analyzed, and the influencing factors of the strategic choices of all entities are explored. This paper uses Matlab 2017 for simulation analysis and provides effective advice on how to urge the government supervision department to effectively supervise, promote the e-commerce platform to enhance review, urge the online seller to legal sale, and encourage consumers feedback which to legally defend their rights.

The remainder of our paper is organized as follows. Section 2 reviews the relevant literature. Section 3 makes assumptions and builds models. The evolutionary stabilization strategies of participants are analyzed in Section 4. In Section 5, the influence of parameters on the results of evolutionary stabilization strategy is simulated and analyzed. Section 6, summarizes the conclusions and points out the direction of future research.

## Literature review

The online shopping method has been widely favored by consumers because of its convenience, but the quality problems of the products that come with it are also criticized. The quality of online shopping products is one of the most popular hot issues in China's e-commerce field, which has aroused widespread concern from the government and the whole society.

Online shopping has made consumer shopping break through the limitations of time and space, but it also brings difficulties to consumer decision-making. An e-commerce environment results in information asymmetry because buyers cannot communicate the quality of products and easily assess the trustworthiness of sellers [3]. Consumers abandon their online purchases at an e-commerce website partly due to the lack of information transparency of the website [4]. In order to retain consumers, online vendors can effectively overcome product-level uncertainty by taking advantage of retailer reputation in the physical world and through the use of digitized video commercials [5]. However, although information asymmetry leads consumers to have less trust in the quality of online shopping products, but there are indeed some consumers who have demand for fake and shoddy products [6]. This consumer demand has created a convergence effect in the online shopping market, which will attract low quality businesses and products into the market [7]. Therefore, with the popularity of online shopping, more and more consumers choose online shopping, and the problem of products quality is also becoming more and more serious.

At present, how to deal with the problem of online shopping products quality has become a hot spot, which has aroused the attention of many scholars. Some scholars have discussed the countermeasures of this problem from the perspective of supply chain. The buyer can induce higher product quality from the risk-neutral supplier than from the risk-averse supplier by conveying the external failures due to returns more effectively to the supplier [8]. Moreover, two common contracts are offered by e-tailers to manufacturers: the revenue sharing contract where a platform appropriates a portion of the manufacturer's revenue, and the fixed fee contract where a platform charges a fixed rent for each sale. Compared with the revenue sharing scheme, the fixed fee scheme leads to a higher price, a higher quality, and a (weakly) smaller market size [9]. It is well known that product quality and environmental conditions fluctuate over time. It is necessary to design a closed-loop supply chain planning framework to deal with such fluctuations [10]. The seller knows most of the information about the product, so reducing seller fraud is critical. Seller-buyer closeness degree-based reputation system can be used to detect fraud products and sellers and elsewhere reduce the fraudulent activities in e-commerce [11]. Supply chain management (SCM) consists of internal practices, which are contained within a firm, and external practices, which cross organizational boundaries integrating a firm with its customers and suppliers. Supplier quality management and customer focus are two QM practices that are also clearly in the domain of SCM [12]. This shows that,

improve the quality of the main body of the supply chain, can effectively improve the quality of products.

However, in a highly competitive market environment, the participating entities are often driven by interests, adopting some negative strategies or even illegal activities, and relying solely on the self-restraint of the participating entities cannot effectively guarantee the quality of online shopping products. Therefore, some scholars believe that in addition to the low level of product quality efforts from sellers and product suppliers, the lack of effective supervision of product quality is also one of the main reasons for online shopping product quality problem. The e-commerce platform plays the most important role and it is the key for the smooth operation of the three-level product quality control system. The e-commerce platform also controls and guides the seller's activities of mitigating product quality uncertainty and improves the quality of products [13]. However, government quality supervision department and e-commerce platforms will be affected by factors such as regulatory costs, penalties, profits, and awards when determining regulatory strategies [14]. Other than this, third-party product evaluation agencies undertake the task of checking and testing product quality in the market, establish the quality level of products, and pass information to consumers [15].

In fact, consumers, as buyers of online shopping green products, play an important role in the online shopping process. When the consumer is dissatisfied with the product, he or she will choose to return it to the seller. Return management can affect the repurchase behavior of consumers, and repurchase behavior will affect the return management process [16]. Consumer attitudes toward online shopping is determined by trust and perceived benefits [17], perceived quality is influenced by the perceptions of competitive price and website reputation, which in turn influences perceived value; perceived value, website reputation, and perceived risk influence online trust, which in turn influence repurchase intention [18]. In addition, consumers as the ultimate experience of online shopping products have the right to feedback on the quality of products, based on consumer evaluation and e-commerce reputation mechanism to study the quality of online shopping products can be an effective complement to existing research. Social control, including consumer evaluation, will inhibit sellers' opportunistic behavior [19]. As a weaker party, consumers should make full use of the online reputation system to make real evaluations and alleviate the losses caused by information asymmetry to the entire consumer group [20]. Considering the profit-seeking nature of each participating subject, collusion problems often occur during the game. The research on collusion is mainly reflected in the networks [21] and pricing [22]. In the field of e-commerce, reputation management systems can be used to reduce the spoofing rating caused by collusion by malicious users [23].

In summary, the existing research focuses on the relationship between government and e-commerce platform or e-commerce platform and online seller. Even though some papers consider multiple participants related to online shopping green product quality supervision, there are few papers that incorporate all parties into the same regulatory system to analyze their relationship with each other, especially those involving conspiracy between participants. At the same time, consumers, as the main body of online shopping, play an important role in the quality supervision of online shopping green products. Most of the existing studies have analyzed the impact of consumers on the reputation construction of the e-commerce platform and the online seller, but have not focused on the role of consumers in the regulatory system. Therefore, this paper considers the loss-sharing relationship between the e-commerce platform and the online seller and the collusion behavior between them. It constructs an evolutionary game model among government supervision department, e-commerce platform and online seller, and introduces consumer feedback in the model. It also analyses the strategic choice and evolutionary trend of participants under different conditions and the indirect

regulatory role of consumers. Finally, using Matlab 2017 to carry out simulation analysis, put forward the countermeasures and suggestions on how to deal with the quality problems of online shopping green products.

## Model assumptions and construction

### Model assumption

The quality supervision of online shopping green products involves multiple stakeholders. This paper takes government supervision department, e-commerce platform, online seller and consumer into account. The game relationship constructed in this paper is shown in Fig 1.

The online seller is the participant 1, the e-commerce platform is the participant 2, and the government supervision department is the participant 3. And assume that all three parties are bounded rational. The online seller's strategy selection space is {provide high quality green products, provide low quality green products}. When the online seller provide low quality green products, in order to enable the products to be listed and circulated on the market to obtain profits, the online seller will have an incentive to collude with the e-commerce platform, that is, bribe the e-commerce platform. The e-commerce platform's strategic selection space is {non-collusion, collusion}, when the e-commerce platform chooses non-collusion, it will neither accept bribes from the online seller offering low quality green products, nor lower the quality review standard; when the e-commerce platform chooses collusion, in addition to accepting bribes from the online seller who provide low quality green products, the e-commerce platform will also lower the quality review standards in order to save costs. The government supervision department's strategy selection space is {strict supervision, loose supervision},when the government supervision department chooses strict supervision, once the collusion is reached, the government supervision department will punish the online seller and the e-commerce platform whether the consumer complaints or not; when the government supervision department chooses loose supervision, only when the consumer complaints, the government supervision department will punish the two parties.

The online seller provides high quality green products with the probability of x, and provides low quality green products with the probability of (1-x); the e-commerce platform chooses non-collusion with the probability of y, and chooses collusion with the probability of

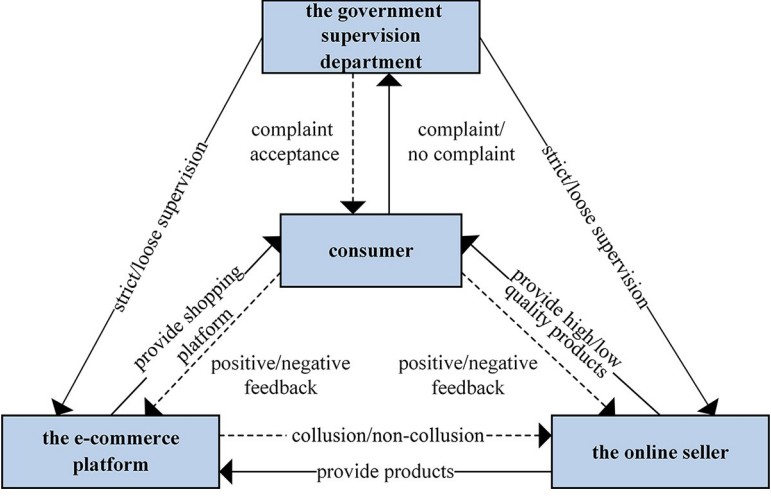

**Fig 1. The game model relationship.**

(1-y); the government supervision department chooses strict supervision with the probability of z, and chooses loose supervision with the probability of (1-z). The cost of providing high quality green products is $C_{sh}$, the cost of providing low quality green products is $C_{sl}$ ($C_{sh} > C_{sl} > 0$). The cost of strict supervision by government supervision department is $C_{gy}$, the cost of loose supervision is $C_{gk}$ ($C_{gy} > C_{gk} > 0$).

Consumer rights will be guaranteed when the online seller provides high quality green products. At this point, consumers will make positive feedback such as praise. This brings additional benefits $E_s$ and $E_p$ to the online seller and e-commerce platform. Conversely, when collusion occurs and low quality green products enter the market, negative feedback (such as bad reviews) will be made by consumers.

When the e-commerce platform chooses non-collusion and actively assumes the responsibility of quality review, the cost of review is $C_{p1}$. At this time, if the online seller provides low quality green products, it will be subject to punishment from the e-commerce platform, the penalty amount is $P$; if the online seller provides high quality green products, the e-commerce platform will give it appropriate rewards. The amount of the reward is $R$.

When the online seller provides low quality green products and the e-commerce platform chooses to collude, the cost of review paid by e-commerce platform is $C_{p2}$, $C_{p2} < C_{p1}$. At this point, the online seller pays the collusion cost $\psi$ and gets the sales income $V$. The collusive income obtained by the e-commerce platform is $\mho$. Low quality green products flow into the market. Consumers will make negative feedback, and the resulting losses $L$ will be shared by the online seller and the e-commerce platform, that is, the losses incurred by the online seller are $\varepsilon L$ ($0 < \varepsilon < 1$), and the losses incurred by the e-commerce platform are $(1 - \varepsilon)L$. At the same time, consumers will choose whether to complain to the government department in order to protect legitimate rights and interests. The probability of complaint is $\delta (0 \leq \delta \leq 1)$, at this time, the government supervision department will punish the online seller and e-commerce platform, and the penalty amounts are $F_s$ and $F_p$ respectively.

## Model construction

Based on the above assumptions, this paper constructs a mixed game matrix between the government supervision department, the e-commerce platform, and the online seller, as shown in Table 1.

## Model analysis

### Analysis of the evolutionary stability strategy of the online seller

The expected profit when the online seller chooses to provide high quality green products is $E_{s1}$.

$$
\begin{aligned}
E_{s1} = {}& yz(V + E_s + R - C_{sh}) + y(1 - z)(V + E_s + R - C_{sh}) + (1 - y)z(V + E_s - C_{sh}) \\
& + (1 - y)(1 - z)(V + E_s - C_{sh}) = V + E_s - C_{sh} + yR
\end{aligned}
\tag{1}
$$

**Table 1. Mixed game matrix for online shopping green product quality supervision.**

| strategy choice | | | the government supervision department | |
|---|---|---|---|---|
| | | | strict supervision $z$ | loose supervision $(1-z)$ |
| the online seller | provide high quality green products $x$ | the e-commerce platform chooses non-collusion $y$ | $V+E_s+R-C_{sh}$ <br> $E_p-R-C_{p1}$, $-C_{gy}$ | $V+E_s+R-C_{sh}$ <br> $E_p-R-C_{p1}$, $-C_{gk}$ |
| | | the e-commerce platform chooses collusion $(1-y)$ | $V+E_s-C_{sh}$ <br> $E_p-C_{p2}$, $-C_{gy}$ | $V+E_s-C_{sh}$ <br> $E_p-C_{p2}$, $-C_{gk}$ |
| | provide low quality green products $(1-x)$ | the e-commerce platform chooses non-collusion $y$ | $-C_{sl}-P$ <br> $P-C_{p1}$, $-C_{gy}$ | $-C_{sl}-P$ <br> $P-C_{p1}$, $-C_{gk}$ |
| | | the e-commerce platform chooses collusion $(1-y)$ | $V-C_{sl}-\psi-\varepsilon L-F_s$ <br> $\mho-(1-\varepsilon)L-F_p-C_{p2}$ <br> $F_s+F_p-C_{gy}$ | $V-C_{sl}-\psi-\varepsilon L-\delta F_s$ <br> $\mho-(1-\varepsilon)L-\delta F_p-C_{p2}$ $\delta F_s+\delta F_p-C_{gk}$ |

The expected profit when the online seller chooses to provide low quality green products is $E_{s2}$.

$$E_{s2} = V - C_{sl} - \psi - \varepsilon L - \delta F_s - y(V + P - \psi - \varepsilon L - \delta F_s) - (1 - y)z(1 - \delta)F_s \quad (2)$$

Therefore, the average expected profit of the online seller is $\bar{E}_s$. $\bar{E}_s = xE_{s1} + (1 - x)E_{s2}$.

The replication dynamic equation for the online seller to choose the "provide high quality green products" strategy ratio is

$$F(x) = dx/dt = x(E_{s1} - \bar{E}_s) = x(x-1)\left\{ \begin{array}{c} C_{sh} - C_{sl} - E_s - \psi - \varepsilon L - \delta F_s - z(1-\delta)F_s \\ -y[V + R + P - \psi - \varepsilon L - \delta F_s - z(1-\delta)F_s] \end{array} \right\} (3)$$

The solutions of $F(x) = 0$ are $x = 0$, $x = 1$, $y = y^* = \frac{C_{sh}-C_{sl}-E_s-\psi-\varepsilon L-\delta F_s-z(1-\delta)F_s}{V+R+P-\psi-\varepsilon L-\delta F_s-z(1-\delta)F_s}$.

Set $m = C_{sh}-C_{sl}-E_s-\psi-\varepsilon L-\delta F_s$, $n = V+R+P-\psi-\varepsilon L-\delta F_s$.

Find partial derivatives for $F(x)$.

$$d(F(x))/dx = (2x - 1)\left\{ \begin{array}{c} C_{sh} - C_{sl} - E_s - \psi - \varepsilon L - \delta F_s - z(1-\delta)F_s \\ -y[V + R + P - \psi - \varepsilon L - \delta F_s - z(1-\delta)F_s] \end{array} \right\} (4)$$

Proposition 1 When $y < y^*$, the evolutionary stability strategy of the online seller is to provide low quality green products; when $y > y^*$, the evolutionary stability strategy of the online seller is to provide high quality green products; when $y = y^*$, there is no evolutionary stability strategy for the online seller.

Proof.

$$\text{Set } K(y) = C_{sh} - C_{sl} - E_s - \psi - \varepsilon L - \delta F_s - y[V + R + P - \psi - \varepsilon L - \delta F_s - z(1-\delta)F_s] \\ - z(1-\delta)F_s. \quad (5)$$

$\partial K(y)/\partial y = -[V+R+P-\psi-\varepsilon L-\delta F_s-z(1-\delta)F_s] < 0$. That is, $K(y)$ is a decreasing function about $y$. If $y = y^*$, $K(y) = 0$, at the this time, there is no evolutionary stability strategy for the online seller. If $y < y^*$, $K(y) > 0$, and $d(F(x))/dx|_{x=0} < 0$, $d(F(x))/dx|_{x=1} > 0$, then providing low quality green products ($x^* = 0$) is the evolutionary stability strategy of the online seller. If $y > y^*$, $K(y) < 0$, and $d(F(x))/dx|_{x=1} < 0$, $d(F(x))/dx|_{x=0} > 0$, then providing high quality green products ($x^* = 1$) is evolutionary stability strategy of the online seller.

Therefore, the evolutionary trend of the online seller's strategy selection is shown in Fig 2.

Let volume $V_{s1}$ represents the probability that the online seller chooses the "provide low quality green products" strategy, and $V_{s2}$ represents the probability of selecting the "provide

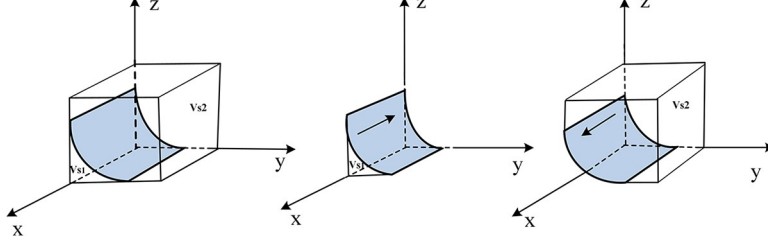

**Fig 2. Evolutionary trend of the online seller strategy selection.**

high quality green products" strategy.

$$V_{s1} = \int_0^1 \int_0^{\frac{m}{(1-\delta)F_s}} \frac{m - z(1-\delta)F_s}{n - z(1-\delta)F_s} dz dx = \frac{1}{(1-\delta)F_s}\left[m - (m-n)\ln\left(1 - \frac{m}{n}\right)\right] \quad (6)$$

$$V_{s2} = 1 - V_{s1} = 1 - \frac{1}{(1-\delta)F_s}\left[m - (m-n)\ln\left(1 - \frac{m}{n}\right)\right] \quad (7)$$

**Corollary 1** The e-commerce platform is more inclined to adopt the "non-collusion" strategy, and the more likely the online seller chooses the "provide high quality products" strategy.

Proof. $y = y^* = (m-z(1-\delta)F_s)/(n-z(1-\delta)F_s)$, If $y<y^*$, $d(F(x))/dx|_{x=0}<0$, $d(F(x))/dx|_{x=1}>0$, providing low quality green products ($x^* = 0$) is the evolutionary stability strategy of the online seller. If $y>y^*$, $d(F(x))/dx|_{x=1}<0$, $d(F(x))/dx|_{x=0}>0$, providing high quality green products ($x^* = 1$) is evolutionary stability strategy of the online seller.

Corollary 1 indicates that if the e-commerce platform is not collusive, it will actively undertake the responsibility of quality review. Once the online seller provides low quality green products, it will be discovered by the strict review system and be punished by the e-commerce platform. The online seller tends to adopt a strategy of providing high quality green products.

**Corollary 2** The increase of additional benefit brought by the positive feedback from the consumers and the increase of loss caused by the negative feedback from the consumers to the online seller can effectively prevent the online seller from providing low quality green products.

Proof. $\partial V_{s1}/\partial E_s = \frac{1}{(1-\delta)F_s}\ln\left(1 - \frac{C_{sh}-C_{sl}-E_s-\psi-\varepsilon L-\delta F_s}{V+R+P-\psi-\varepsilon L-\delta F_s}\right) < 0$, $V_{s1}$ is a decreasing function for $E_s$, and $V_{s1}$ decreases as $E_s$ increases, which the increase in additional profit of positive feedback from consumers will reduce the probability of low quality green products offered by the online seller. $\partial V_{s1}/\partial L = -\varepsilon m/n(1-\delta)F_s<0$, $V_{s1}$ is a decreasing function for $L$, and $V_{s1}$ decreases as $L$ increases. That is to increase the losses caused by negative feedback from consumers, which will reduce the probability of the online seller providing low quality green products.

Corollary 2 shows that consumers will give feedback on online shopping green products such as online praise or bad reviews. This kind of feedback will affect the reputation of the online seller, and will also influence the decision-making of other buyers, thus becoming a factor of particular concern for the online seller. The increase in the amount of the additional profit brought by positive feedback from consumers and the losses caused by negative feedback from consumers to the online seller will have a significant impact on the strategic choice of the online seller. In order to establish a good reputation and maintain long-term development, the online seller will consciously reduce the probability of providing low quality green products.

**Corollary 3** The e-commerce platform establishes a strict review system to increase the amount of incentives for providing high quality green products to the online seller and the amount of penalties for providing low quality green products, which can prevent low quality products from being offered by the online seller.

Proof. $\frac{\partial V_{s1}}{\partial R} = \frac{1}{(1-\delta)F_s}\left[\ln\left(1 - \frac{C_{sh}-C_{sl}-E_s-\psi-\varepsilon L-\delta F_s}{V+R+P-\psi-\varepsilon L-\delta F_s}\right) + \frac{C_{sh}-C_{sl}-E_s-\psi-\varepsilon L-\delta F_s}{V+R+P-\psi-\varepsilon L-\delta F_s}\right] < 0$, $V_{s1}$ is a decreasing function for $R$, and $V_{s1}$ decreases as $R$ increases. The e-commerce platform increase incentives for the online seller of high quality green products, which can reduce the probability of providing low quality green products.

$\frac{\partial V_{s1}}{\partial P} = \frac{1}{(1-\delta)F_s}\left[\ln\left(1 - \frac{C_{sh}-C_{sl}-E_s-\psi-\varepsilon L-\delta F_s}{V+R+P-\psi-\varepsilon L-\delta F_s}\right) + \frac{C_{sh}-C_{sl}-E_s-\psi-\varepsilon L-\delta F_s}{V+R+P-\psi-\varepsilon L-\delta F_s}\right] < 0$, $V_{s1}$ is a decreasing function for $P$, and $V_{s1}$ decreases as $P$ increases, which increase the penalty for the online seller who

providing low quality green products of the e-commerce platform can improve the deterrent effect for the online seller, so that the online seller can choose the strategy of providing high quality green products actively in order to avoid suffering high penalties.

Corollary 3 indicates that during the online shopping process, the e-commerce platform bears the responsibility of third-party credit and quality supervision, and the establishment of a strict review system for the e-commerce platform is crucial for green product quality supervision. But the online seller can effectively avoid the punishment from the e-commerce platform by means of modern advanced technology. Therefore, in addition to the strict punishment system, the e-commerce platform can increase the incentive amount for the credited online sellers, in order to stop the online seller from providing low quality green products.

**Corollary 4** The government supervision department can smooth complaint channels and increase the probability of complaints from consumers, so as to reduce the probability of low quality green products offered by the online seller.

Proof. $\frac{\partial V_{s1}}{\partial \delta} = -\frac{C_{sh}-C_{sl}-E_s-\psi-\varepsilon L-\delta F_s}{(1-\delta)(V+R+P-\psi-\varepsilon L-\delta F_s)} < 0$, $V_{s1}$ is a decreasing function for $\delta$, and $V_{s1}$ decreases as $\delta$ increases, which increases the probability of consumer complaints will reduce the probability of low quality green products offered by the online seller.

Corollary 4 shows that the government supervision department should encourage consumers to participate in quality supervision work together, cultivate their awareness of safeguarding their legitimate rights and interests, and increase the probability of complaints to government supervision department. That is, consumer complaints play an indirect role in regulating the quality of online shopping green products. In order to gain long-term development, maintain a good reputation and evade punishment from the government, the online seller will avoid offering low quality green products.

## Analysis of the evolutionary stability strategy of the e-commerce platform

The expected profit when the e-commerce platform chooses "non-collusion" is $E_{p1}$.

$$E_{p1} = x(E_p - R - P) + P - C_{p1} \tag{8}$$

The expected profit when the e-commerce platform chooses "collusion" is $E_{p2}$.

$$E_{p2} = x(E_p - C_{p2}) - (1-x)(1-\delta)zF_p + (1-x)[\text{Ʊ} - (1-\varepsilon)L - \delta F_p - C_{p2}] \tag{9}$$

Therefore, the average expected profit of the e-commerce platform is $\bar{E}_p$,
$\bar{E}_p = yE_{p1} + (1-y)E_{p2}$.

The replication dynamic equation for the e-commerce platform to choose the "non-collusion" strategy ratio is:

$$F(y) = dy/dt = y(E_{p1} - \bar{E}_p)$$
$$= y(y-1)\left\{ \begin{array}{l} \text{Ʊ} - P - (1-\varepsilon)L - \delta F_p - (1-x)z(1-\delta)F_p \\ -x[\text{Ʊ} - R - P - (1-\varepsilon)L - \delta F_p] + C_{p1} - C_{p2} \end{array} \right\} \tag{10}$$

The solutions of $F(y) = 0$ are

$$y = 0, \ y = 1, \ z = z^*$$
$$= \frac{\text{Ʊ} - P - (1-\varepsilon)L - \delta F_p + C_{p1} - C_{p2} - x[\text{Ʊ} - R - P - (1-\varepsilon)L - \delta F_p]}{(1-x)(1-\delta)F_p}.$$

Find partial derivatives for $F(y)$.

$$d(F(y))/dy = (2y-1)\left\{\begin{array}{l} \mho - P - (1-\varepsilon)L - \delta F_p - (1-x)z(1-\delta)F_p \\ -x[\mho - R - P - (1-\varepsilon)L - \delta F_p] + C_{p1} - C_{p2} \end{array}\right\} \quad (11)$$

Proposition 2 When $z<z^*$, the evolutionary stability strategy of the e-commerce platform is collusion; when $z>z^*$, the evolutionary stability strategy of the e-commerce platform is non-collusion; when $z = z^*$, there is no evolutionary stability strategy of the e-commerce platform.

Proof.

$$\text{Set } Q(z) = \mho - P - (1-\varepsilon)L - \delta F_p + C_{p1} - C_{p2} - (1-x)z(1-\delta)F_p - x[\mho - R - P - (1-\varepsilon)L - \delta F_p]. \quad (12)$$

$\partial Q(z)/\partial z = -(1-x)(1-\delta)F_p<0$. That is, $Q(z)$ is a decreasing function about $z$.

If $z = z^*$, $Q(z) = 0$, at the this time, there is no evolutionary stability strategy for the e-commerce platform. If $z<z^*$, $Q(z)>0$ and $d(F(y))/dy|_{y=0}<0$, $d(F(y))/dy|_{y=1}>0$, then collusion ($y^* = 1$) is evolutionary stability strategy of the e-commerce platform. If $z>z^*$, $Q(z)<0$ and $d(F(y))/dy|_{y=1}<0$, $d(F(y))/dy|_{y=0}>0$, then non-collusion ($y^* = 1$) is evolutionary stability strategy of the e-commerce platform.

Therefore, the evolutionary trend of the e-commerce platform's strategy selection is shown in Fig 3.

Let volume $V_{p1}$ represents the probability that the e-commerce platform chooses the "collusion "strategy, and $V_{p2}$ represents the probability of selecting the" non-collusion "strategy.

Set $a = \mho - P(1-\varepsilon)L - \delta F_p + C_{p1} - C_{p2}$, $b = \mho - R - P - (1-\varepsilon)L - \delta F_p$.

$$V_{p1} = \int_0^1 \int_0^1 \frac{a-bx}{(1-x)(1-\delta)F_p}dxdy = \frac{1}{(1-\delta)F_p}\left[b + (a-b)\int_0^1 \frac{1}{1-x}dx\right] \quad (13)$$

$$V_{p2} = 1 - \int_0^1 \int_0^1 \frac{a-bx}{(1-x)(1-\delta)F_p}dxdy = 1 - \frac{1}{(1-\delta)F_p}\left[b + (a-b)\int_0^1 \frac{1}{1-x}dx\right] \quad (14)$$

**Corollary 5** The probability that the e-commerce platform chooses a non-collusion strategy will rise as the probability of the government supervision department selecting a strict supervision strategy rises.

Proof. Make $z = z^* = (a-bx)/(1-x)(1-\delta)F_p$. If $z<z^*$ and $d(F(y))/dy|_{y=0}<0$, $d(F(y))/dy|_{y=1}>0$, then collusion ($y^* = 0$) is evolutionary stability strategy of the e-commerce platform. If $z>z^*$ and $d(F(y))/dy|_{y=1}<0$, $d(F(y))/dy|_{y=0}>0$, then non-collusion ($y^* = 1$) is evolutionary stability strategy of the e-commerce platform.

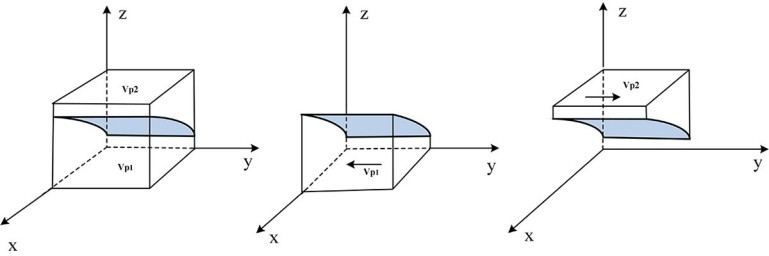

**Fig 3. Evolutionary trend of the e-commerce platform strategy selection.**

Corollary 5 shows that when the government supervision department choose strict supervision, it is more likely to discover illegal behavior. Once the collusion is discovered, strict punishment will be imposed on the e-commerce platform. Therefore, the e-commerce platform tends to choose non-collusion strategy in order to avoid penalties.

**Corollary 6** The impact of consumer complaints on the choice of the e-commerce platform depends on the government supervision department's penalty for the e-commerce platform.

Proof. $\partial V_{p2}/\partial \delta = ((1-\delta)F_p - (b + (a-b)\int_0^1 \frac{1}{1-x}dx))/(1-\delta)^2 F_p$

When $(b + (a-b)\int_0^1 \frac{1}{1-x}dx)/(1-\delta) > F_p$, $\partial V_{p2}/\partial \delta < 0$, $V_{p2}$ is a decreasing function about $\delta$. $V_{p2}$ decreases as $\delta$ increases. That is to increase the probability of complaints from consumers, and the probability of e-commerce platform choosing non-collusion is decreased.

When $(b + (a-b)\int_0^1 \frac{1}{1-x}dx)/(1-\delta) < F_p$, $\partial V_{p2}/\partial \delta > 0$, $V_{p2}$ is an increasing function about $\delta$. $V_{p2}$ increases as $\delta$ increases. That is to increase the probability of complaints from consumers, and will promote the choice of non-collusion of e-commerce platform.

Corollary 6 indicates that consumer complaints can constrain the e-commerce platform, but its effectiveness depends on the government's implementation of the complaint results, that is, whether to impose severe penalties on the e-commerce platform. When the penalty amount is low and the threshold value is not reached, even if the probability of consumer complaints continues to increase, the penalty amount is still within the acceptable range of the e-commerce platform, which is an unbelievable threat to the e-commerce platform. Plunging the speculative psychology of e-commerce platforms, the probability of choosing collusion is higher. When the government supervision department's penalty amount is greater than the threshold, it is enough to form a deterrent effect on the e-commerce platform. Once the consumer makes a complaint, the amount of punishment imposed by the e-commerce platform will be huge. At this time, the increase in the probability of consumer complaints can play a role in constraining the e-commerce platform. That is, as the probability of consumer complaints increases, the probability of e-commerce platforms choosing non-collusion strategy is gradually increasing.

**Corollary 7** As the e-commerce platform and the online seller share responsibility and share risks, and increase the amount of losses caused by negative feedback from consumers, the amount of losses shared by the e-commerce platform also increases, which promotes the e-commerce platform to choose non-collusion strategy.

Proof. $\partial V_{p2}/\partial L = (1-\varepsilon)/(1-\delta)F_p > 0$. $V_{p2}$ is an increasing function about $L$. $V_{p2}$ increases as $L$ increases. That is the increase in losses caused by negative feedback from consumers can effectively guide the e-commerce platform to choose non-collusion strategy for maintaining reputation.

Corollary 7 shows that the strengthening of the relationship between the e-commerce platform and the online seller can promote mutual restraint, and form a situation of glory and loss. Under this constraint, the e-commerce platform will tend to adopt non-collusion strategy in order to protect its own interests.

### Analysis of the evolutionary stability strategy of the government supervision department

The expected profit when the government supervision department chooses "strict supervision" is $E_{g1}$.

$$E_{g1} = F_s + F_p - C_{gy} - x(F_s + F_p) - (1-x)y(F_s + F_p) \tag{15}$$

The expected profit when the government supervision department chooses "loose supervision" is $E_{g2}$.

$$E_{g2} = \delta F_s + \delta F_p - C_{gk} - x(\delta F_s + \delta F_p) - (1-x)y(\delta F_s + \delta F_p) \tag{16}$$

Therefore, the average expected profit of the government supervision department is $\bar{E}_g$, $\bar{E}_g = zE_{g1} + (1-z)E_{g2}$.

The replication dynamic equation for the government supervision department to choose the "strict supervision" strategy ratio is:

$$F(z) = dz/dt = z(E_{g1} - \bar{E}_g)$$
$$= z(z-1)[C_{gy} - C_{gk} - (1-y)(1-\delta)(F_s + F_p) + x(1-y)(1-\delta)(F_s + F_p)] \tag{17}$$

The solutions of $F(z) = 0$ are $z = 0$, $z = 1$, $x = x^* = \frac{(1-y)(1-\delta)(F_s+F_p)+C_{gk}-C_{gy}}{(1-y)(1-\delta)(F_s+F_p)}$.

Find partial derivatives for $F(z)$.

$$d(F(z))/dz = (2z-1)[C_{gy} - C_{gk} - (1-y)(1-\delta)(F_s + F_p) + x(1-y)(1-\delta)(F_s + F_p)] \tag{18}$$

Proposition 3 When $x<x^*$, the evolutionary stabilization strategy of the government department is strict supervision; when $x>x^*$, the evolutionary stability strategy of the government department is loose supervision; when $x = x^*$, there is no evolutionary stability strategy of the government department.

Proof.

$$\text{Set } D(x) = C_{gy} - C_{gk} - (1-y)(1-\delta)(F_s + F_p) + x(1-y)(1-\delta)(F_s + F_p). \tag{19}$$

$\partial D(x)/\partial x = (1-y)(1-\delta)(F_s+F_p)>0$. That is, $D(x)$ is an increasing function about $x$. If $x = x^*$, $D(x) = 0$, at the this time, there is no evolutionary stability strategy for the government supervision department. If $x<x^*$, $D(x)<0$, and $d(F(z))/dz|_{z=1}<0$, $d(F(z))/dz|_{z=0}>0$, then strict supervision ($z^* = 1$) is evolutionary stability strategy of the government supervision department. If $x>x^*$, $D(x)>0$, $d(F(z))/dz|_{z=1}>0$, $d(F(z))/dz|_{z=0}<0$, then loose supervision ($z^* = 0$) is evolutionary stability strategy of the government supervision department.

Therefore, the evolutionary trend of the government supervision department's strategy selection is shown in Fig 4.

Let volume $V_{g1}$ represents the probability that the government supervision department chooses the" strict supervision "strategy, and $V_{g2}$ represents the probability of selecting the" loose supervision "strategy.

Set $u = (1-\delta)(F_s+F_p)$, $v = (1-\delta)(F_s+F_p)+C_{gk}-C_{gy}$.

$$V_{g1} = \int_0^1 \int_0^{\frac{v}{u}} \frac{v-uy}{(1-y)u}dydz = \frac{v}{u} + \left(1-\frac{v}{u}\right)\ln\left(1-\frac{v}{u}\right) \tag{20}$$

$$V_{g2} = 1 - \int_0^1 \int_0^{\frac{v}{u}} \frac{v-uy}{(1-y)u}dydz = 1 - \left[\frac{v}{u} + \left(1-\frac{v}{u}\right)\ln\left(1-\frac{v}{u}\right)\right] \tag{21}$$

**Corollary 8** As the probability of the online seller choosing to provide high quality green product and the probability of the e-commerce platform selecting non-collusion strategy increase, the probability that the government supervision department chooses the strict supervision strategy will decrease.

Proof. Make $x = x^* = (v-uy)/(1-y)u$. If $x<x^*$, $d(F(z))/dz|_{z=1}<0$, $d(F(z))/dz|_{z=0}>0$, then strict supervision ($z^* = 1$) is evolutionary stability strategy of the government supervision

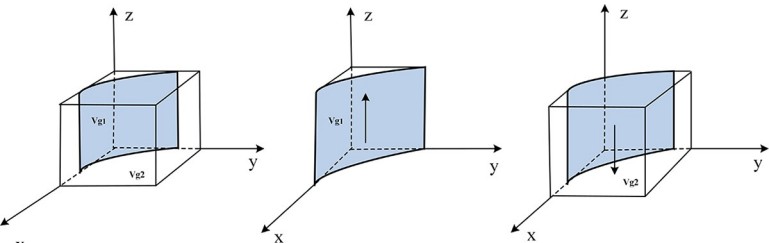

**Fig 4. Evolutionary trend of the government supervision department strategy selection.**

department. If $x > x^*$, $d(F(z))/dz|_{z=1} > 0$, $d(F(z))/dz|_{z=0} < 0$, then loose supervision ($z^* = 0$) is evolutionary stability strategy of the government supervision department.

Make $y = y^* = (v - ux)/(1 - x)u$. If $y < y^*$, $d(F(z))/dz|_{z=1} < 0$, $d(F(z))/dz|_{z=0} > 0$, then strict supervision ($z^* = 1$) is evolutionary stability strategy of the government supervision department. If $y > y^*$, $d(F(z))/dz|_{z=1} > 0$, $d(F(z))/dz|_{z=0} < 0$, then loose supervision($z^* = 0$) is evolutionary stability strategy of the government supervision department.

Corollary 8 indicates that the greater the probability that the online seller chooses to provide high quality green products, the greater the possibility that the market can achieve good operation by relying solely on the self-discipline of the transaction subject. The government supervision department tends to deregulate in order to save supervision costs and adopts loose supervision.

The e-commerce platform is the main body of quality review, which controls whether the green products can be put on the shelves and flow into the market. The higher the probability that the e-commerce platform chooses non-collusion, the higher the possibility of undertaking the review responsibility and prohibiting the listing of low quality green products. The government supervision department tends to trust the e-commerce platform and decentralize its supervision responsibilities.

**Corollary 9** The higher the probability that consumer will complain after his or her own rights have been compromised, the lower the probability that the government supervision department will strictly supervise.

Proof. $\partial V_{g1}/\partial \delta = (C_{gy} - C_{gk})((1 - \delta)(F_s + F_p))\ln(1 - ((1 - \delta)(F_s + F_p) + C_{gk} - C_{gy})/(1 - \delta)(F_s + F_p)) < 0$, $V_{g1}$ is a decreasing function about $\delta$. $V_{g1}$ decreases as $\delta$ increases. That is, the increase in the probability of consumer complaints will prompt government supervision department to loose supervision.

Corollary 9 indicates that there is a certain substitution effect between consumers' complaints and the supervision of government department, that is, both can restrain and deter the online seller and the e-commerce platform. Consumer complaints play an indirect supervision role in the quality of green products. When the probability of complaints from consumers is high, the probability of strict supervision by the government supervision department is reduced, which not only saves supervision costs, but also exerts constraints on the online seller and the e-commerce platform.

## Simulation analysis

In order to more intuitively reflect the influence of various parameters under the collusion on the game equilibrium and equilibrium strategy between the government supervision department, the e-commerce platform and the online seller, we will use Matlab 2017 software to simulate the impact of each parameter on the results of evolutionary stability strategy.

Set $C_{sh} = 16, C_{sl} = 14, E_s = 8, \psi = 6, \varepsilon = 0.6, V = 20, \mho = 6, C_{p1} = 12, C_{p2} = 5, C_{gy} = 10, C_{gk} = 8$.

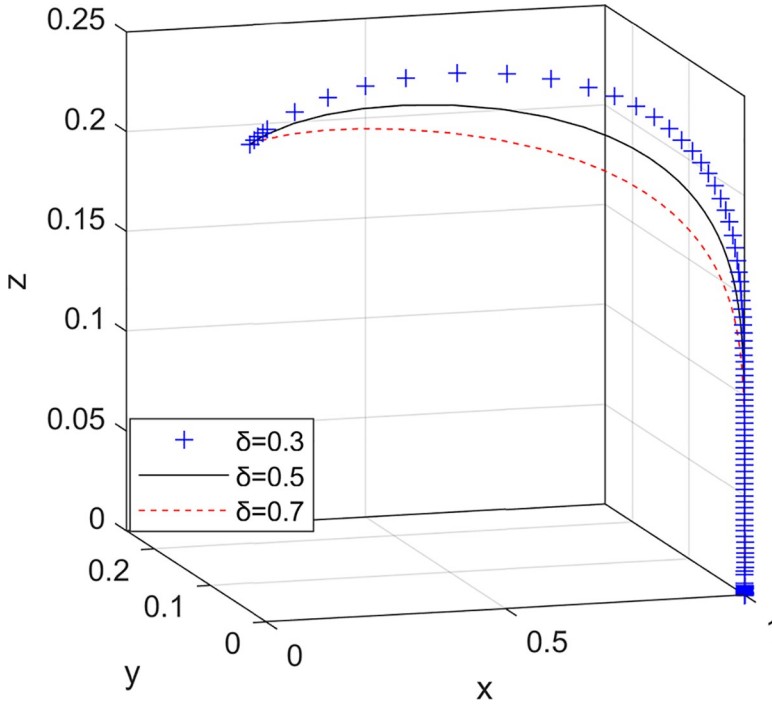

**Fig 5. The impact of consumer complaint probability on the choice of tripartite strategy.**

## The impact of consumer complaint probability

The probability of consumer complaints reflects the level of consumer awareness of rights protection and the degree of participation in the supervision of online shopping quality. Now consider: the consumer complaint mechanism is constantly improving, the cost of complaints is reduced; the processing time for complaints is shortened; the government supervision department can implement a strict punishment system based on consumer complaints. That is to say, under the above conditions, the probability of complaints from consumers after purchasing low quality green products is gradually increased. Set $\delta = \{0.3, 0.5, 0.7\}$.

Fig 5 shows that the probability of the online seller providing high quality green products has gradually increased, and the rate of stability in providing high quality green products has accelerated with the increase in the probability of consumer complaints, and the government supervision department and the e-commerce platform have adjusted accordingly: Once the online seller is stable in providing high quality green products, it will has no incentive to bribe the e-commerce platform. In order to save the review cost, the e-commerce platform tends to choose a collusion strategy. At this time, the collusion cannot be achieved, so the collusion strategy of the e-commerce platform only represents that it lowers the standard for review; because the online seller are subject to consumer complaints and tend to proactively provide high quality green products, the government supervision department will reduce their interference in the online seller and the e-commerce platform and reduce the probability of strict supervision.

## The impact of consumer negative feedback

The e-commerce platform provides consumers with a place for online shopping. At the same time, as a third-party supervision, in order to protect the legitimate rights and interests of consumers, it is responsible for the strict examination of the green products provided by the online

seller. Once a consumer purchases a low quality green product, it not only makes a negative evaluation of the online seller, but also the e-commerce platform will be negatively affected to some extent due to the review of dereliction of duty. Therefore, the losses $L$ caused by negative feedback are shared by the e-commerce platform and the online seller. Now consider: the degree of consumers' active participation in evaluation is enhanced. After purchasing low quality green products, they will actively make real evaluations of the online seller and e-commerce platform, that is, the losses caused by negative feedback gradually increase. Set $L = \{10,15,20\}$.

Fig 6 shows that the probability of the online seller chooses to provide high quality green products is increasing. And the larger $L$, the faster the online seller is stable at providing high quality green products. When the probability of the online seller providing high quality green products is increasing, the probability of non-collusion of the e-commerce platform and the probability of strict supervision of the government department begin to decrease. Finally, the online seller is stable in providing high quality green products, the e-commerce platform is stable in collusion strategies, and the government supervision department is stable in loose supervision. This is because once the online seller is stable at providing high quality green products, it can guarantee the quality of the green products itself, so the e-commerce platform and the government supervision department do not need to monitor the online seller. At this time, the e-commerce platform tends to collusion, and the collusion strategy only represents that it lowers the standard for review, the government supervision department tends to loose supervision.

## The impact of reward and punishment of the e-commerce platform on the choice of the online seller

In order to undertake the responsibility of quality review and strict quality control, the e-commerce platform will implement a strict reward and punishment system for the online seller.

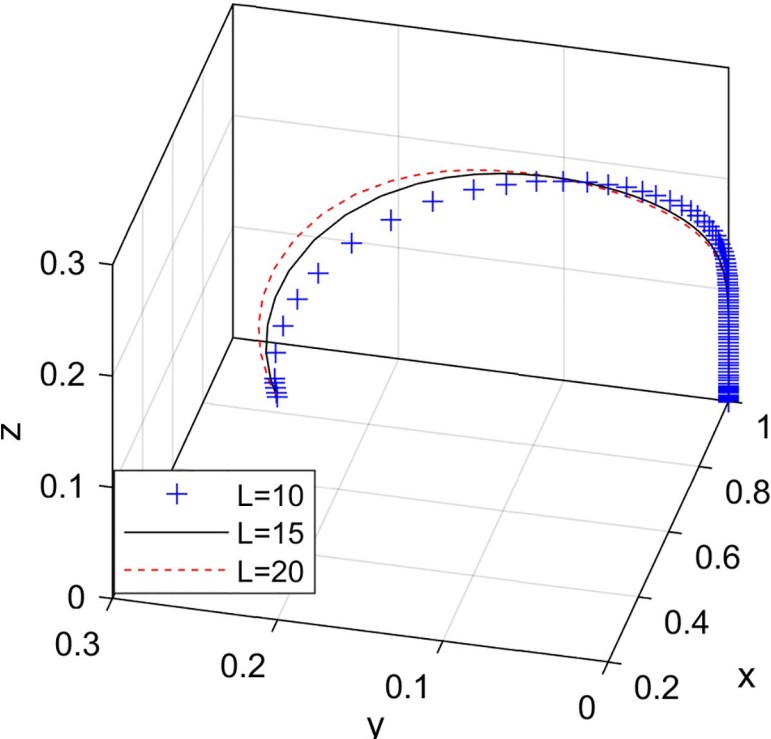

**Fig 6. The impact of negative feedback on the choice of tripartite strategy.**

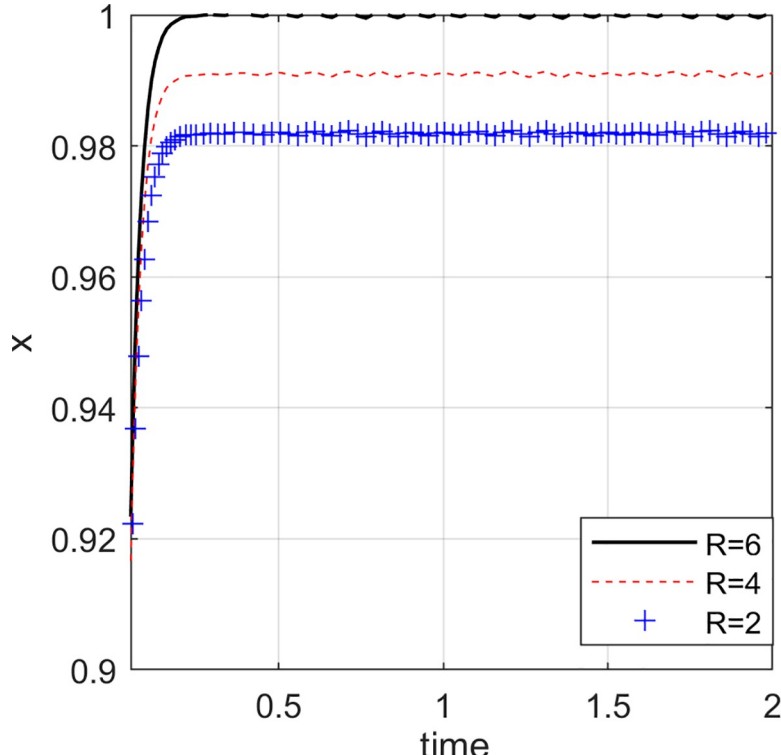

**Fig 7. Impact of e-commerce platform incentives.**

That is, the online seller is appropriately rewarded and punished by the e-commerce platform. As the sense of responsibility of the e-commerce platform has gradually increased, the reward and punishment system has become stricter, and the penalty $P$ and reward $R$ for the online seller have gradually increased. Set $y = 0.5$, $z = 0.5$, $R = \{2,4,6\}$, $P = \{0,2,7\}$.

Fig 7 and Fig 8 show that as the e-commerce platform increases the incentives and penalties for the online seller, the probability of the online seller choosing to provide high quality green products is increasing. If the e-commerce platform chooses non-collusion and actively assumes the responsibility of quality review, it will impose strict penalties on online seller offering low quality green products, therefore, the online seller will tend to provide high quality green products. The higher the incentive for the e-commerce platform to the online seller, the more attractive it is to the online seller, and the probability of choosing high quality green products bigger.

### The impact of government supervision department penalties

As the public's attention to quality issues and the supervision of the government supervision department gradually increased, the penalty for the online seller and the e-commerce platform increased. Set $y = 0.5$, $z = 0.5$, $F_s = \{3,7,9\}$ or $x = 0.5$, $z = 0.5$, $F_p = \{2,3,4\}$.

Fig 9 shows that with the increase in the amount of penalty for government supervision department, the probability of the online seller choosing to provide high quality green products and the probability of the e-commerce platform choosing non-collusion will gradually increase. The government supervision department punish stricter for the online seller and the e-commerce platform, which will have the stronger deterrent effect. In order to avoid excessive

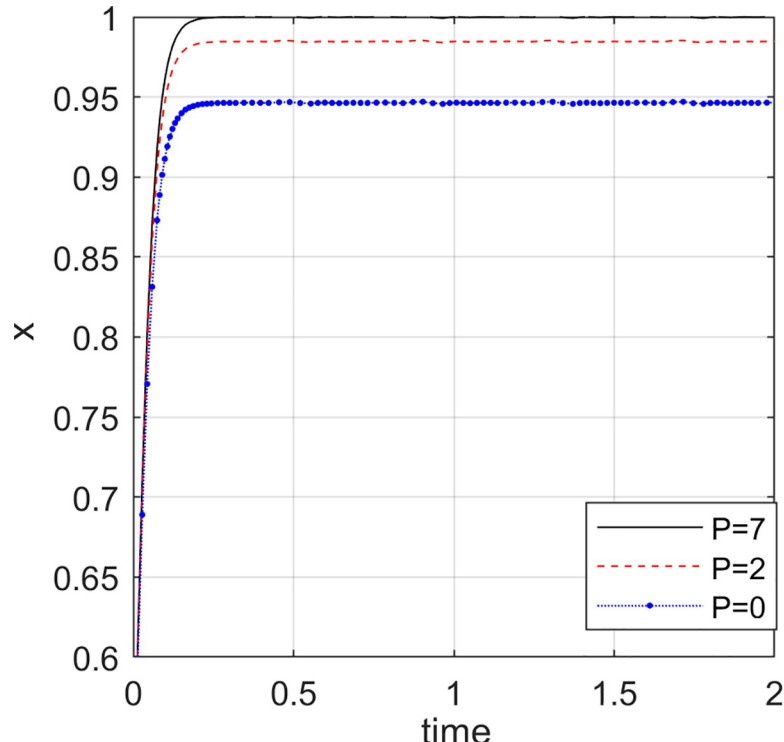

**Fig 8. Impact of e-commerce platform penalties.**

punishment, both parties tend to operate or trade legally, that is, the online seller provide high quality green products, and the e-commerce platform is not collusion.

## Conclusions and future works

Combined with the above game model analysis and simulation analysis, the following conclusions and recommendations are drawn.

Consumer complaints play an indirect role in the supervision of online sellers. Consumers can reasonably and legally use the major social networking sites, shopping platforms, etc. to evaluate the real shopping experience. However, the role of consumer complaints depends on the implementation of the results of complaints by the government supervision department, that is, the punishment of the relevant subjects. Therefore, the government should encourage consumers to complain by reducing the cost of complaints, shortening the time of accepting complaints and implementing the results of complaints.

The stronger the relationship between the online sellers and e-commerce platforms, the more they can share the responsibility for the quality review and legal operation. In order to enhance the connection between the two, on the one hand, government departments should establish a sound joint responsibility system. On the other hand, government departments should encourage consumers to make reasonable and realistic evaluations through various modern network technologies, thereby strengthening the mutually constrained relationship between e-commerce platforms and online sellers, that is, if the online sellers illegally sell low quality green products, the reputation of the e-commerce platform is also damaged, and the customer usage rate of the platform is reduced, which in turn affects the sales of the online sellers.

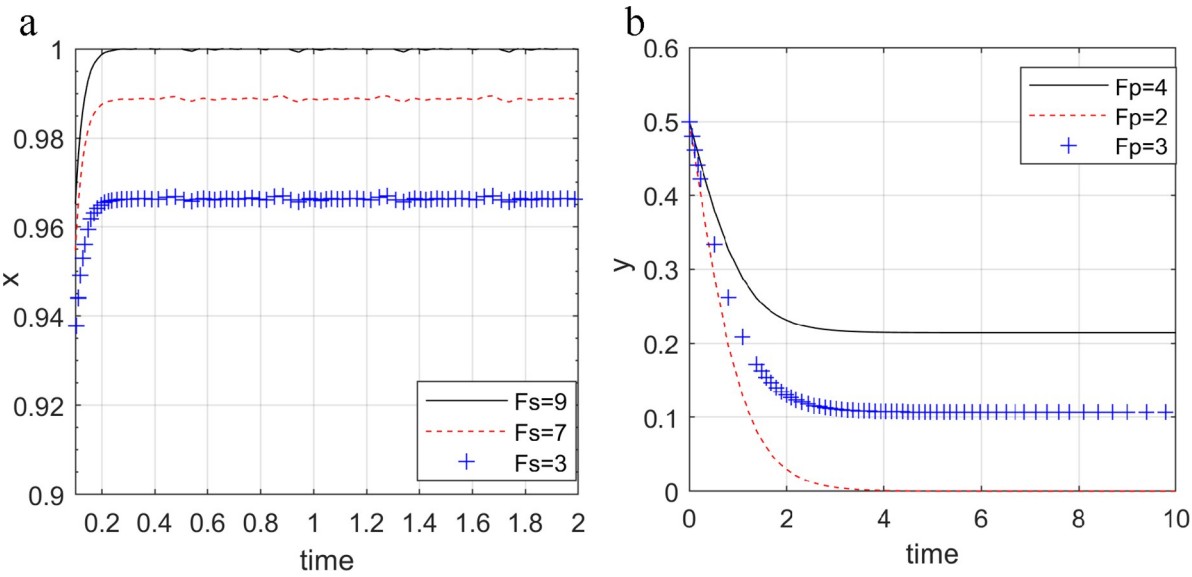

**Fig 9.** Impact of government penalties on the online seller (a) and the e-commerce platform (b).

Only the punishment system can't fully exert the binding effect of the e-commerce platform on the online sellers, and it is also crucial to establish an incentive system for the e-commerce platform. Since the e-commerce platform cannot obtain enough quality information of the green products, in order to prevent the illegal sales of the online seller, a strict punishment system should be established. Such as reducing their credit scores, prohibiting the sale of products, etc. However, modern technology is developed, and the online sellers will use various means to avoid punishment or make up for the damage caused by punishment, such as hiring others to make a single order to improve credibility. Therefore, the e-commerce platform must also adopt some positive incentives such as improving credit scores and reducing advertising costs.

Government supervision department plays an important role in quality supervision. First, government departments should establish a sound green product certification mechanism. Second, the government supervision department should actively respond to the relevant policies of the state, assume the responsibility of strict supervision in accordance with the newly promulgated "E-commerce Law", severely punish violations of laws and regulations in the course of online transactions and timely exposure of relevant events to expand its scope of influence. In addition, market regulation is carried out to promote reasonable competition between e-commerce platforms to guide them to pay attention to green product quality issues.

The popularity of online shopping has also brought many green product quality problems, and the quality supervision involves multiple parties. This paper considers the collusion behavior and consumer feedback of the online seller and the e-commerce platform, with the government supervision department, the online seller, and the e-commerce platform as the main players. Constructing incomplete information, dynamic and repetitive evolutionary game models involving government supervision departments, online sellers, e-commerce platforms and product suppliers will be the next research direction.

## Acknowledgments

The authors are grateful to the referees for their valuable comments and their helps on how to improve the quality of our paper.

## Author Contributions

**Writing – original draft:** Hui He.

**Writing – review & editing:** Lilong Zhu.

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
