## [Decision Letter · Decision Letter 0]

2 Jan 2020

PONE-D-19-32986

Online Shopping Green Product Quality Supervision Strategy with Consumer Feedback and Collusion Behavior

PLOS ONE

Dear Dr. Zhu,

Thank you for submitting your manuscript to PLOS ONE. After careful consideration, we feel that it has merit but does not fully meet PLOS ONE’s publication criteria as it currently stands. Therefore, we invite you to submit a revised version of the manuscript that addresses the points raised during the review process.

We recommend that it should be revised taking into account the changes requested by the reviewers. Since the requested changes includes Major Revision, the revised manuscript will undergo the next round of review by the same reviewers.

We would appreciate receiving your revised manuscript by Feb 16 2020 11:59PM. To enhance the reproducibility of your results, we recommend that if applicable you deposit your laboratory protocols in protocols.io, where a protocol can be assigned its own identifier (DOI) such that it can be cited independently in the future. For instructions see: http://journals.plos.org/plosone/s/submission-guidelines#loc-laboratory-protocols

We look forward to receiving your revised manuscript.

Kind regards,

Baogui Xin, Ph.D.

Academic Editor

PLOS ONE

Journal Requirements:

The authors are grateful to the referees for their valuable comments and their helps on how to

improve the quality of our paper. This work was supported by the Humanities and Social Sciences

Foundation of the Ministry of Education in China under grant No.17YJA630147, Nature Science

Foundation of Shandong Province under grant No.ZR2019MG017 and National Social Science

Foundation of China under grant No.13AGL012.

No

3.  We suggest you thoroughly copyedit your abstract section for language usage, spelling, and grammar. If you do not know anyone who can help you do this, you may wish to consider employing a professional scientific editing service.  

4. Please include your tables as part of your main manuscript and remove the individual files. Please note that supplementary tables (should remain/ be uploaded) as separate "supporting information" files.

No

Reviewers' comments:

Reviewer's Responses to Questions

**Comments to the Author**

1. Is the manuscript technically sound, and do the data support the conclusions?

Reviewer #1: Yes

Reviewer #2: Yes

Reviewer #3: Yes

Reviewer #4: Yes

2. Has the statistical analysis been performed appropriately and rigorously? 

Reviewer #1: Yes

Reviewer #2: Yes

Reviewer #3: Yes

Reviewer #4: N/A

3. Have the authors made all data underlying the findings in their manuscript fully available?

Reviewer #1: Yes

Reviewer #2: Yes

Reviewer #3: Yes

Reviewer #4: No

4. Is the manuscript presented in an intelligible fashion and written in standard English?

Reviewer #1: Yes

Reviewer #2: Yes

Reviewer #3: Yes

Reviewer #4: No

5. Review Comments to the Author

Reviewer #1: I think the structure of the manuscript is clear. It describes a technically reliable scientific research, and can draw a more appropriate and rigorous analysis based on the derivation results. The biggest innovation of the manuscript is to include the consumers as the subject in the evolutionary game model, and explore the role of consumer feedback in the regulatory system. However I think there are still a few errors in the manuscript.

Firstly, equation (16) on page 14 should be “The expected profit when the government supervision department chooses loose supervision”, but in the manuscript it is “The expected profit when the government supervision department chooses strict supervision”.

Secondly, first three lines of the first paragraph on page 17 , ”Figure 5 shows that with the increase in the probability of consumer complaints, the probability of strict supervision by government departments is gradually increasing, the probability of non-collusion of e-commerce platform is gradually decreasing.” I think it should be ” Figure 5 shows that with the increase in the probability of consumer complaints, the probability of strict supervision by government departments is gradually increasing, the probability of collusion of e-commerce platform is gradually decreasing.”. I do not know if it is my misunderstanding or the author accidentally wrote it wrong.

Thirdly, last paragraph on page 17, ”Until the online seller stabilized and chose to provide high quality green products, the government's strategic choices gradually stabilized over loose supervision, and the e-commerce platform strategy choice gradually stabilized against collusion.……Once the strategy choice of online seller is stable in providing high quality green products, the e-commerce platform and the government supervision department do not need to monitor the online seller, and they can guarantee the quality of the green products themselves. At this time, the e-commerce platform tends to collusion, the government supervision department tends to loose supervision.” I think these two sentences are inconsistent, I hope the author can check them further.

Finally, I would like to make a suggestion. I think in the manuscript the statement of strict supervision and loose supervision of the government supervision department is not particularly clear. I think the authors can clearly point out that when the government supervision department chooses strict supervision, regardless of whether the consumer complains, the government department will punish the online seller and the e-commerce platform ; when the government department chooses loose supervision, only when the consumer complains, the government department will punish the two parties . I think in this way it will be easier for readers to understand this part clearly.

---

## [Author Response · Author response to Decision Letter 0]

6 Feb 2020

Responses to Editor and Reviewers

Dear Editor Professor Baogui Xin and Reviewers,

Thank you very much for giving us an opportunity to submit a revised version of the manuscript entitled “Online Shopping Green Product Quality Supervision Strategy with Consumer Feedback and Collusion Behavior.” The manuscript is revised submission (Manuscript ID: PONE-D-19-32986) with new line and page numbers in the text, some grammar and spelling errors had also been corrected. The revised paper contains 24 pages, 9 figures and 1 table. 

We are also thankful to the reviewers for their critical reading and valuable comments on the manuscript. Those comments were very helpful for providing direction for our further studies. The relevant changes had been made in the original manuscript according to the comments of reviewers, and the major revised portions were marked in red. We also responded point by point to each reviewer comments as listed below, along with a clear indication of the location of the revision. In addition, after careful consideration, we have further improved the paper without changing the structure and general content of the paper, and marked it with a bold blue font.

Responds to the editor’s comments:

Comment #1:

When submitting your revision, we need you to address these additional requirements. Please ensure that your manuscript meets PLOS ONE's style requirements, including those for file naming. 

Response: We have carefully read the requirements of the journal and modified the manuscript to make it meets PLOS ONE's style requirements, including those for file naming. 

Comment #2:

We note that you have provided funding information that is not currently declared in your Funding Statement. However, funding information should not appear in the Acknowledgments section or other areas of your manuscript. We will only publish funding information present in the Funding Statement section of the online submission form. Please remove any funding-related text from the manuscript and let us know how you would like to update your Funding Statement. Currently, your Funding Statement reads as follows: No.

Response: Funding information has been removed from the acknowledgment section and does not appear in any part of the manuscript. And we have updated the funding information in the Funding Statement section of the online submission form according to journal requirements. In the Cover Letter, we declare that the author received specific funding for this work, which was supported by the Humanities and Social Sciences Foundation of the Ministry of Education in China under grant No.17YJA630147, Nature Science Foundation of Shandong Province under grant No.ZR2019MG017 and National Social Science Foundation of China under grant No.13AGL012.

Comment #3: 

We suggest you thoroughly copyedit your abstract section for language usage, spelling, and grammar. If you do not know anyone who can help you do this, you may wish to consider employing a professional scientific editing service. 

Response: Thank you very much for the suggestions. The manuscript does have some sentence structure and grammatical errors. We have used professional scientific editing service to correct the language usage, spelling, and grammar. This Manuscript’s English has been edited by Wiley English Language Editing Services (http://wileyeditingservices.com/en/).

Comment #4: 

Please include your tables as part of your main manuscript and remove the individual files. Please note that supplementary tables (should remain/ be uploaded) as separate "supporting information" files.

Response: We make sure that tables are included as part of main manuscript and the individual files are removed. And all the data are within the manuscript, there are no supporting information files.

Comment #5:

Please complete your Competing Interests on the online submission form to state any Competing Interests. If you have no competing interests, please state "The authors have declared that no competing interests exist.” as detailed online in our guide for authors at http://journals.plos.org/plosone/s/submit-now.

Response: We have no competing interests, and have stated "The authors have declared that no competing interests exist." as detailed online in guide for authors at http://journals.plos.org/plosone/s/submit-now.

Responds to the reviewer’s comments:

Firstly, equation (16) on page 14 should be “The expected profit when the government supervision department chooses loose supervision”, but in the manuscript it is “The expected profit when the government supervision department chooses strict supervision.”

Response: Thank you very much for the suggestions. Yes, there are indeed errors here, we are very sorry for our incorrect writing. Equation (16) on page 14 (page 15 after modification) should really be “The expected profit when the government supervision department chooses loose supervision,” and we have made correction according to the reviewers’ comments. 

Secondly, first three lines of the first paragraph on page 17 , ”Fig 5 shows that with the increase in the probability of consumer complaints, the probability of strict supervision by government departments is gradually increasing, the probability of non-collusion of e-commerce platform is gradually decreasing.” I think it should be ” Figure 5 shows that with the increase in the probability of consumer complaints, the probability of strict supervision by government departments is gradually increasing, the probability of collusion of e-commerce platform is gradually decreasing.”. I do not know if it is my misunderstanding or the author accidentally wrote it wrong.

Response: Thank you very much for the suggestions. Yes, I agree with your opinion. First three lines of the first paragraph on page 17 ( page 18 after modification ) are not clear or accurate, we have re-written this part according to the reviewers’ comments and made a detailed statement on the strategy choices of the e-commerce platform and the government supervision department in the model assumption and construction part. In addition, in order to conduct model analysis more clearly, this article further proposes three propositions based on the original manuscript. When the e-commerce platform chooses non-collusion, it will neither accept bribes from the online seller offering low quality green products, nor lower the quality review standard; when the e-commerce platform chooses collusion, in addition to accepting bribes from the online seller who provide low quality green products, the e-commerce platform will also lower the quality review standards in order to save costs. At the same time, when the government supervision department chooses strict supervision, once the collusion is reached, the government supervision department will punish the online seller and the e-commerce platform whether the consumer complaints or not; when the government supervision department chooses loose supervision, only when the consumer complaints, the government supervision department will punish the two parties .Therefore, Fig 5 shows that with the increase in the probability of consumer complaints, the probability of the online seller providing high quality green products has gradually increased, and the rate of stability in providing high quality green products has accelerated, and the government supervision department and the e-commerce platform have adjusted accordingly: Once the online seller is stable in providing high quality green products, it will has no incentive to bribe the e-commerce platform. At this time, in order to save the review cost, the e-commerce platform tends to choose a collusion strategy. At this time, the collusion cannot be achieved, so the collusion strategy of the e-commerce platform only represents that it lowers the standard for review; because the online seller are subject to consumer complaints and tend to proactively provide high quality green products, the government supervision department will reduce their interference in the online seller and the e-commerce platform and reduce the probability of strict supervision.”

Thirdly, last paragraph on page 17, ”Until the online seller stabilized and chose to provide high quality green products, the government's strategic choices gradually stabilized over loose supervision, and the e-commerce platform strategy choice gradually stabilized against collusion.……Once the strategy choice of online seller is stable in providing high quality green products, the e-commerce platform and the government supervision department do not need to monitor the online seller, and they can guarantee the quality of the green products themselves. At this time, the e-commerce platform tends to collusion, the government supervision department tends to loose supervision.” I think these two sentences are inconsistent, I hope the author can check them further. 

Response: Thank you very much for the suggestions. Considering the reviewers’ suggestion, we have modified the last paragraph on page 17 (page 19 after modification). We are so sorry for the ambiguity caused by our writing errors. The previous manuscript used the wrong prepositions, which changed the meaning of the paper, so we edited this paragraph again and elaborated more, we have corrected as “Fig 6 shows that with the increase in the loss caused by negative feedback, the probability of the online seller choose to provide high quality green products is increasing. And the larger , the faster the online seller is stable at providing high quality green products. When the probability of the online seller providing high quality green products is increasing, the probability of non-collusion of the e-commerce platform and the probability of strict supervision of the government department begin to decrease. Finally, the online seller is stable in providing high quality green products, the e-commerce platform is stable in collusion strategies, and the government supervision department is stable in loose supervision. This is because once the online seller is stable at providing high quality green products, it can guarantee the quality of the green products itself, the e-commerce platform and the government supervision department do not need to monitor the online seller, at this time, the e-commerce platform tends to collusion, and the collusion strategy only represents that it lowers the standard for review, the government supervision department tends to loose supervision.”

Finally, I would like to make a suggestion. I think in the manuscript the statement of strict supervision and loose supervision of the government supervision department is not particularly clear. I think the authors can clearly point out that when the government supervision department chooses strict supervision, regardless of whether the consumer complaints, the government department will punish the online seller and the e-commerce platform; when the government department chooses loose supervision, only when the consumer complaints, the government department will punish the two parties. I think in this way it will be easier for readers to understand this part clearly.

Response: Thank you very much for the suggestions. It is really true as reviewer suggested that the statement of strict supervision and loose supervision of the government supervision department should be clear. Therefore, in the hypothesis part of the model, we have described the two strategic choices of the government supervision department in more detail and more clearly. We have corrected as “When the government supervision department chooses strict supervision, once the collusion is reached, the government supervision department will punish the online seller and the e-commerce platform whether the consumer complaints or not; when the government supervision department chooses loose supervision, only when the consumer complaints, the government supervision department will punish the two parties.”

We have tried our best to improve the manuscript and made some substantial changes and necessary deletions according to the editors’ and reviewers’ comments. We earnestly appreciate the editors’ and reviewers’ professional work and hope that the corrections will make our manuscript suitable for publication in PLOS ONE. We are looking forward to receiving comments from reviewers in the future. If you have any questions, please do not hesitate to contact me at the address below.

Once again, thank you very much for your valuable comments and suggestions.

Best wishes.

Sincerely,

Lilong Zhu, Ph.D. and Professor. Postdoctoral research in Shandong University. Research scholar in College of business, University of Illinois at Urbana-Champaign, USA. Ph.D. degree was granted in management science and engineering from Tongji University and Postdoctoral research in Shandong University. His research interests are supply chain management and quality management.

E-mail: zhulilong2008@126.com Tel: +86-13853193366

College of Business, Shandong Normal University, School of Management, Shandong University, Ji’nan 250014, Shandong, China.

---

## [Editor Report · Decision Letter 1]

7 Feb 2020

Online Shopping Green Product Quality Supervision Strategy with Consumer Feedback and Collusion Behavior

PONE-D-19-32986R1

Dear Dr. Zhu,

We are pleased to inform you that your manuscript has been judged scientifically suitable for publication and will be formally accepted for publication once it complies with all outstanding technical requirements.

With kind regards,

Baogui Xin, Ph.D.

Academic Editor

PLOS ONE
---

## [Editor Report · Acceptance letter]

24 Feb 2020

PONE-D-19-32986R1 

Online Shopping Green Product Quality Supervision Strategy with Consumer Feedback and Collusion Behavior 

Dear Dr. Zhu:

I am pleased to inform you that your manuscript has been deemed suitable for publication in PLOS ONE. Congratulations! Your manuscript is now with our production department. 

With kind regards,

on behalf of

Prof. Baogui Xin 

Academic Editor

PLOS ONE